# Preparation of a Novel Fracturing Fluid System with Excellent Elasticity and Low Friction

**DOI:** 10.3390/polym11101539

**Published:** 2019-09-20

**Authors:** Yang Zhang, Jincheng Mao, Jinzhou Zhao, Tao Xu, Anqi Du, Zhaoyang Zhang, Wenlong Zhang, Shaoyun Ma

**Affiliations:** 1State Key Laboratory of Oil and Gas Reservoir Geology and Exploitation, Southwest Petroleum University, Chengdu 610500, China; yangzhang10234@163.com (Y.Z.); 15827742134@163.com (T.X.); 13880147377@163.com (A.D.); 201711000107@stu.swpu.edu.cn (W.Z.); 2Shanghai King Materials Industry Limited Liability Company, Shanghai 201700, China; msy08007@163.com

**Keywords:** hydrophobic associative water-soluble polymers, fracturing fluid, excellent elasticity and low friction, mechanism analysis, thermodynamic property

## Abstract

The quaternary polymer was synthesized by radical polymerization and characterized by ^1^H NMR. The tests of critical associating concentration and SEM suggest that there is a multilayered and robust network structure in the polymer solution. An excellent elasticity in the polymer solution by the first normal stress difference, viscoelasticity, and thixotropy was observed. The critical crosslinker concentration of polymer with sodium dodecyl sulfate and its interaction mechanism were investigated. According to the reaction kinetics, the supramolecular structure had the lowest activation energy, stable network structure, and greater thermal stability. Then the polymer was employed in the fracturing fluid due to its excellent elasticity using the intermolecular forces, which showed superior sand suspension capacity by dynamic sand suspension measurement. Meanwhile, a theoretical analysis was proposed as to why polymer solution has excellent suspension and drag reduction properties. Therefore, this polymer could be an alternative in many fields, especially in fracking, which is significant for the development of oil and gas resources in deep wells.

## 1. Introduction

Fracturing fluid is typically employed to reconstruct oil and gas reservoirs in hydraulic fracturing (commonly referred to in the media as “fracking”) [1]. Fracturing fluids play an important role in forming cracks, transferring pressure, and carrying proppant into fractures during the operation of hydraulic fracturing [2]. A fluid’s performance will determine the success or failure of the hydraulic fracturing operation in a given area, as well as the magnitude to which its production is increased.

Typically, the type of fracturing fluid thickener used in an oilfield can be divided into guar gum or its derivatives. Examples include hydroxypropyl guar gum (HPG), viscoelastic surfactants, and hydrophobic associating water-soluble polymer (HAWSP) [3]. However, guar gum-based fracturing fluids usually have the disadvantages of high insoluble residue, poor shear resistance, and large damage to reservoirs due to the plugging of pore throats, which would restrict the application in the low permeability reservoirs [4,5]. In contrast to this, viscoelastic surfactant-based fracturing fluids will typically rely on molecular self-assembly to form micelles for transporting proppants into cracks [6,7]. However, viscoelastic surfactants are not widely used in oil fields due to their serious loss of filtration, the difficulty of breaking the gel, and their high costs [8]. HAWSP, a synthetic polymer, can be prepared for specific functions by changing the polymerization monomers or the polymerization method [9,10]. In fact, it has been found that HAWSP in fracking can play an effective role in carrying sand via its own spatial network structures in aqueous solution [11,12,13]. However, it is well known that the initial viscosity of most HAWSP-based fracturing fluids is too large due to its special structure, which will increase the friction between the fracturing fluid and pipeline. In other words, an extra burden for fracturing equipment would be presented, which would increase the risk of equipment failure over time.

Meanwhile, some scholars have proposed that leveraging the benefits of elasticity outweighs the advantages offered from increasing the viscosity of fracturing fluids when it comes to improving sand-carrying capacity. Therefore, a fracturing fluid with high elasticity and low viscosity should be developed to improve its sand transport capacity and decrease the friction between fracturing fluid and pipeline [14,15,16].

Based on the achievements of our group and the pioneer works in this field in recent years in polymer-based fracturing fluids [8,9,10,13,17], it is believed that the elasticity of polymer solutions can now be efficiently improved by utilizing hydrophobic association interactions, electrostatic bridges effects, hydrogen bonds, and van der Waals forces. Moreover, note that increasing the elasticity of a polymer solution also works to reduce the friction caused by pumping the fluid, as the energy released in turbulent flow can be effectively absorbed and converted into the elastic potential energy of the molecular chain instead [18]. With these considerations in mind, naturally, a quaternary polymer was designed in our group by copolymerizing acrylamide (AM), acrylic acid (AA), 4-isopropenylcarbamoyl-benzene sulfonic acid (AMBS), and *N*-(3-methacrylamidopropyl)-*N*,*N*-dimethyldodecan-1-aminium (DM-12), which was named HELV because the polymer solution exhibited high elasticity and low viscosity. Obviously, there are three intermolecular forces in the solution, which contribute to an increase in the elasticity of the HELV solution: (1) electrostatic bridges between DM-12 and AMBS groups can be formed in solution; (2) H-bonds formed between water and amide would strengthen the binding forces between the polymer chains; (3) the long-chain hydrophobic monomer would further increase the binding forces due to the hydrophobic interaction. In addition, the initial viscosity of HELV solution was reduced through optimizing the molecular weight of HELV. Therefore, a copolymer solution with low viscosity as well as high elasticity was developed by introducing a benzene ring, sulfonates, and long hydrophobic chains in the polymer structure. 

In other words, this technical provided a good answer to the dilemma between increasing the viscosity of fracturing fluid to improve the proppant suspension capacity and large friction between liquids and pipelines. Therefore, it is interesting and significant to investigate fracturing fluid which could be an alternative for the development of oil and gas resources in deep wells.

## 2. Experimental Section

### 2.1. Materials

Acrylamide (AM, CAS: 79-06-1), acrylic acid (AA, CAS: 79-10-7), sodium hydroxide (NaOH, CAS: 1310-73-2), sodium dodecyl sulfate (SDS, CAS: 751-21-3), and 2,2′-azobis (2-methylpropionamide) dihydrochloride (V50, CAS: 2997-92-4) were all purchased from Chengdu Kelong Chemical Reagents Corporation (Chengdu, China). 4-Isopropenylcarbamoyl-benzene sulfonic acid (AMBS) and *N*-(3-methacrylamidopropyl)-*N*,*N*-dimethyldodecan-1-aminium (DM-12) were prepared in our lab as functional monomers. All chemicals and reagents were utilized without further purification. Ammonium persulfate (APS, CAS: 7727-54-0), hydroxypropyl guar gum (HPG), and proppant were applied by GuangHan Innovative Technology Development Institute (Deyang, China). Deionized water was obtained from a water purification system. All chemicals were analytical reagent grade and were utilized without further purification.

### 2.2. Synthesis of HELV

Appropriate amounts of acrylamide (12.0 g), acrylic acid (3.0 g), 4-isopropenylcarbamoyl-benzene sulfonic acid (0.4 g), and *N*-(3-methacrylamidopropyl)-*N*,*N*-dimethyldodecan-1-aminium (0.4 g) were placed in deionized water, the total monomer concentration was maintained at 30 wt%. Subsequently, the pH of solution was adjusted to 7.0 using NaOH. Then, 2,2′-azobis (2-methylpropionamide) dihydrochloride solution was added using a syringe; the amount of V50 was 0.07 wt% of the total mass of the monomers. Meanwhile, the solution was placed in a UV light fixture (T5 8W UVB) at 25 °C. Polymer colloid was prepared via illumination reaction after 8 h. This copolymer was named “HELV”. The structural formulae for the process of synthesizing HELV is presented in Scheme 1. Thereafter, the polymer was cut into small pieces. It was then purified by using ethanol precipitation three times. Lastly, the compound was dried in a vacuum desiccator and stored for further use.

### 2.3. Critical Association Concentration

For this experiment, the viscosities of solutions with the addition of 0.01~0.3 wt% HELV were measured by the HAAKE MARS III (006-1322) rheometer (Haake, Karlsruhe, Germany) under a shear rate of 7.34 s^−1^ and a temperature of 25 °C. The relationship curve between viscosity and concentration was acquired thereafter. The point at which viscosity mutation occurred was concluded to be the point of critical association concentration.

### 2.4. Shear Sensitivity Testing

In addition, 50 mL of the polymer aqueous solution was taken in a measuring cylinder and was placed into the HAAKE MARS III (006-1322) rheometer (Haake, Karlsruhe, Germany). Then, the polymer aqueous solution was subjected to continuous shear testing with a shear rate ranging from 1 to 200 s^−1^ under 25 °C for 30 min using the HAAKE MARS III (006-1322) rheometer (Haake, Karlsruhe, Germany). The viscosity retention rate was calculated by the viscosity at 200 s^−1^ divided by the viscosity at 100 s^−1^.

### 2.5. Viscoelasticity Measurements

Viscoelasticity evaluation was carried out via dynamic rheological experimentation using oscillation measurements. Stress and frequency for the HELV solution were determined by dynamic stress sweep as well as frequency sweep measurements using an Anton PPar rheometer (MR302, Anton Paar, Austria) with CP50-1-SN30644 plate fixture (diameter = 0.099 mm). All of these experiments were carried out using stress-controlled testing. To ensure consistency in experimental conditions, all samples were measured at 25 °C.

### 2.6. Critical Crosslinking Concentration

For viscoelastic fluids, the tangent of loss angle (tan δ = *G″*/*G′*) can be characterized by the fluid state of the solution in question. Based on past research in the field [19], it has been suggested that critical concentration can be determined by plotting a solution’s concentration against its tan δ at variable frequencies. For this study, solutions with different amounts of HELV were crosslinked with 0.015 wt% sodium dodecyl sulfate (SDS) solution. The tangent of loss angle was then measured using the HAAKE MARS III (006-1322) rheometer (Haake, Karlsruhe, Germany) under different frequencies at 25 °C. The concentration at which all relationship curves were found to have the same value of tan δ was designated as the critical crosslinking concentration (*C*_cc_). 

### 2.7. Thixotropy

The formation and destruction of HELV structures were characterized by thixotropy. Correspondingly, the areas between the upgoing and downgoing curves were found to reflect this thixotropy. For this experiment, the systems containing HELV and SDS were measured using a plate PP50 by adjusting the three parameters on the rotary mode of the HAAKE MARS III (006-1322). For the first stage, it was found that the shear rate rises from 0 to 100 s^−1^. For the second stage, it was found that the shear rate maintains itself at 0 for 50 s. For the third stage, it was found that the shear rate drops from 100 to 0 s^−1^.

### 2.8. Viscoelastic Heat-Resistance

Measurements for viscoelastic heat-resistance were conducted using the Anton Paar rheometer (MCR302, Anton Paar, Austria). The fracturing fluids were measured at a sinusoidal amplitude of 0.1% with an oscillation frequency of 1 Hz. For this experiment, the temperature was raised from 25 to 80 °C over a period of 30 min. In addition, samples were put under a temperature of 80 °C in order to measure the viscoelastic heat resistance of the fluids further.

### 2.9. Dynamic sand Suspension

Dynamic sand suspension was evaluated through visual inspection using a slab crack simulation. A diagram of the study’s simulation system is presented in Figure 1. Testing was conducted at 25 °C. Specifically, for this experiment the dimensions of the plate crack were 4000 mm × 8 mm × 300 mm. The proppants were gauged at 20/40 mesh and 30/50 mesh and subsequently mixed with the study’s fracturing fluids. The sand radio used here was 20%. In the last step, the mixed fluid was pumped into the slab with 1 m^3^/min displacement.

### 2.10. Field-Scale Friction Reduction Evaluation

A circulating pipe system was used to test for drag reduction. Clean water was used to calibrate the system, and different drag reduction solutions were tested under variable conditions thereafter. A 6 mm pipeline was used to test the relationship between pressure drop and flow rate. Specifically, drag reduction is defined by Equation (1), as follows [12]:(1)$$\eta =\frac{\Delta P_{0}-\Delta P_{1}}{\Delta P_{0}}\times 100\%$$
where *η* represents drag reduction in HELV, ∆*P*_0_ represents the pressure drop from clean water (MPa), and ∆*P*_1_ represents the pressure drop from drag reduction agent HELV (MPa).

## 3. Results and Discussion

### 3.1. The ^1^H NMR of HELV

Figure 2 presents the spectrum results for HELV using a ^1^H nuclear magnetic resonance spectrometer (NMR) (400 MHz, D_2_O). The proton signals at 4.70 ppm were assigned to the solvent protons (D_2_O). The proton signals at 1.57–1.68 ppm (a) and 2.14–2.25 ppm (b) were attributed to the CH_2_–CH– in the polymer main chain. The proton signals at 0.93–0.96 ppm (c) could be assigned to the –CH_3_ protons in the polymer main chain. The proton signals at 3.72–3.78 ppm (d), 3.37–3.42 ppm (f) and 1.19–1.26 ppm (g) were associated with the –CH_2_–CH_2_ protons in DM-12. The proton signals at 3.10–3.16 ppm (e) were assigned to the N(CH_3_)_2_ on the DM-12. The proton signals at 0.77–0.79 ppm (h) were from the –CH_3_ in polymer side chains. The proton signals at 6.78–6.80 ppm (j) and 7.50–7.52 ppm (i) were due to the benzene ring in the AMBS. The proton signals at 1.09–1.12 ppm and 3.55–3.60 ppm were associated with the –CH_3_ and the –CH_2_ in ethanol. The proton signals at 5.56–6.21 ppm (k) were due to the vinyl at the residual monomers. All results verified that the synthesized polymers were consistent with the targets.

### 3.2. Critical Association Concentration

When a water-soluble polymer is hydrated, the viscosity of the polymer solution under a shear rate of 7.34 s^−1^ will increase with concentration. In such a case, the viscosity will appear to exhibit an exponential increase as the solution’s concentration rises to a certain value. This value is referred to as the solution’s critical association concentration.

As shown in Figure 3, it was found that there are two obvious shifts at the concentration points of 270 and 1800 mg/L. This is due to the special structure of polymers containing the hydrophobic long chain and benzene rings, which is more complicated than the monotonous upward trend of the viscosity of conventional polymers, such as guar gum. Naturally, this curve was divided into three phases by these two points. These two points are labeled hereafter as the first critical associating concentration (*C_1_^*^*) and second critical associating concentration (*C_2_^*^*), respectively. 

It was found that there are fewer polymer macromolecules in solution when the concentration is less than *C_1_^*^*, exhibiting a reduced number of intramolecular association microregions formed by the hydrophobic interaction between the polymer solution’s hydrophobic groups. Subsequently, when the polymer solution’s concentration lies between *C_1_^*^* and *C_2_^*^*, a large number of hydrophobic microdomains are exhibited. During this stage, the solution’s macromolecular chains will become stretched, and hydrodynamic volume will increase significantly as a result. This interaction works to increase the viscosity of the polymer solution dramatically. It can be seen that a supramolecular network structure is formed in the solution here via hydrophobic interaction, electrostatic interaction, and hydrogen bonding. Moving on to when the concentration exceeds *C_2_^*^*, intermolecular entanglement between macromolecules will begin to occur, with a multilayered and robust spatial network structure in the aqueous solution. As a result, the volume of fluid mechanics will increase significantly, which elicits a sharp increase in the polymer solution’s viscosity [20]. The association process for the copolymer solution is presented in Figure 4. It can be concluded that the polymer solution is characterized by a robust spatial network structure, at least to a certain extent.

### 3.3. Shear Sensitivity Test

The variable shear test is one of the primary factors affecting the success or failure of a hydraulic fracturing operation. With this in mind, all of the study’s test fluids were analyzed rheologically regarding both their viscous and elastic properties. Figure 5 presents the shear viscosity versus shear rate profiles for the fluids tested in this study. Within the observed range of shear rate (from 1 to 200 s^−1^), all fluids were found to exhibit shear-thinning as their shear viscosities decreased corresponding with increasing shear rate. It is clear that the copolymer fluid containing 0.3 wt% HELV was found to have the highest shear viscosity, while the solution containing 0.1 wt% HELV exhibited the lowest shear viscosity. The fluid containing 0.2 wt% HELV had a shear viscosity lying between these two extremes. Moreover, compared the viscosity of the fluids at 100 s^−1^, the retention rates of the viscosity of all of the test fluids were 69.06%, 67.37%, and 66.35% at 200 s^−1^, respectively. These results indicate that the HELV solution has good shear resistance, indicating that the introduction of hydrophobic long chains and benzene ring would prevent the bending of the polymer backbone.

Consistency index (κ) values as well as flow behavior index (η) values for the tested fluids were obtained by curve fitting the power law model to the study’s shear stress versus shear rate data (Equation (2)): (2)lgτ=ηlgγ+lgκ
where *τ* represents the shear stress (Pa), *η* represents the rheological index, *γ* represents the shear rate (s^−1^), and *κ* represents the consistency coefficient (mPa·s^n^).

The fitting curve for the shear stress versus shear rate profiles is presented in Figure 6. Note that the data provided in Table 1 shows that all of the three tested fluids exhibited shear-thinning characteristics. The flow behavior index values were found to be nearly the same for the tested fluids. However, the values for the consistency index of the fluid containing 0.3 wt% HELV was found to be greater than that of those for the fluid with 0.2 wt% HELV, followed by the fluid with 0.1 wt% HELV. In addition, the fluid containing 0.3 wt% HELV was also found to be 1.35 times more viscous than the fluid containing 0.2 wt% HELV and 1.7 times more viscous than the fluid containing 0.1 wt% HELV.

### 3.4. First Normal Stress Difference N_1_

During initial hydraulic fracturing operations, the fracturing fluid in a pipe will be characterized by a turbulent flow state. The flow state of fracturing fluid during the processes of a first normal stress difference test will be similar to its flow state in a pipeline. As such, a first normal stress difference test can serve to describe the non-linear viscoelastic behavior of fracturing fluid in the process of flow.

On the subject of shear flow, the polymer fluid is found to exhibit a strange elastic behavior in addition to its aforementioned viscosity. This indicates a normal stress difference related to the nonlinear effect. The magnitude of first normal stress difference is a measure of the elastic effect of a polymer fluid, which may exhibit behavior characterized by the power law model within a certain range of shear rate [21]:
*N*_1_ = *Ar^m^*(3)
where *A* and *m* are all constants for the test liquid, 1 < *m* < 2. The variable *N*_1_ represents the first normal stress difference and *r* denotes the shear rate.

The first normal stress difference curves for the study’s HELV solution are presented in Figure 7. It was found that the first normal stress difference changes little when the shear rate is less than 100 s^−1^. However, the difference increases rapidly at a shear rate from 100 to 1000 s^−1^. Additionally, the value of the first normal stress difference *N*_1_ of the solution increased with the increase of HELV concentration under the same shear stress condition. Note that this dynamic explains why the spatial network structures formed by intermolecular hydrophobic association within the HELV solution will resist deformation when the fluid is sheared. 

### 3.5. Viscoelasticity Measurement

In order to study the length of relaxation time in the study’s fluids, it is necessary to measure for a frequency sweep through oscillatory testing. Relative results are presented in Figure 8a–c. It is clear that the storage modulus (*G′*) and loss modulus (*G″*) will increase as angular frequency increases. With the further increase of scanning frequency, the storage modulus is greater than the loss modulus when the scanning frequency is greater than the critical frequency, at which point curves *G′* and *G″* cross each other (*G*_c_), which suggests that the storage modulus plays a dominant role in this time. The relaxation time *t*_c_ corresponding to the crossing point *G*_c_ can be used to describe the solution viscoelasticity. A longer *t*_c_ verifies that the solution structure makes a greater contribution to elastic efficiency. According to the calculation results, the relaxation times of different solutions are shown in Table 2. The study’s tested fluids exhibited matching longest relaxation time at 0.60 s, with the exception of the fluid containing 0.1 wt% HELV, which exhibited a time of 0.18 s. Based on detailed analysis of the experimental observations and data, it can be concluded that the concentration for HELV is less than the second critical associating concentration (*C_2_^*^*) when the HELV concentration of the solution is 0.1 wt%. Therefore, there are only small amounts of hydrophobic microdomains in the polymer solution at this time. Macroscopically, the viscosity and elasticity of the resulting fluid will be relatively low.

Meanwhile, it is found that there is a robust spatial network structure in the polymer solution from a microscopic point of view. It is the reason that the effects of electrostatic bridges, H-bonds, and hydrophobic association, strengthen the stability of the spatial network structure. Thus, it is proved that the HELV solution has excellent viscoelasticity from a microscopic point of view.

### 3.6. Critical Crosslinker Concentration

For this experiment, SDS was crosslinked with the study’s HELV solution. The crosslinking mechanisms of HELV with SDS are presented in Figure 9. Mixed micelles were formed from the interaction between the hydrophobic chains of the polymer and the hydrophobic heads of the surfactant. These formations result in polymer chains that tend toward unfolding and associating with the surfactant [22]. Moreover, larger and stronger hydrophobic microregions will form from crosslinking, which works to increase viscosity and elasticity in the fluid.

As noted earlier, it has been suggested in past research that a solution’s critical concentration can be determined by plotting the tangent of loss angle (tan δ = *G″*/*G′*) versus its concentration over variable frequencies. The concentration at which all of the relationships curves are found to have the same value for tan δ will be the solution’s critical crosslinking concentration (*C*_cc_) [19]. For this study’s testing for *C*_cc_, the results are presented in Figure 10. The critical crosslinking concentration for HELV is found to be at 0.155 wt%, indicating that the critical crosslinking concentration for HELV is less than that for HPG (0.285 wt%) [19]. These findings indicate that a fracturing fluid with a small amount of HELV can greatly improve viscoelasticity. Simply put, this liquid can decrease the friction that commonly occurs between fracturing fluid and pipelines, thereby reducing the damage inflicted on a reservoir during operation.

### 3.7. Thixotropy

The thixotropy of the study’s solutions with 0.1, 0.2 and 0.3 wt% of HELV and 0.015 wt% of SDS were measured at 25 °C. The results are presented in Figure 11. It is clear that these three studies’ solutions all formed a viscoelastic ring at a lower shear rate, indicating the presence of a network structure within the solution and the ability to store a portion of the energy during the shearing process. It is worth emphasizing that the upgoing and the downgoing lines of the 0.1 wt% solution observed in the figure nearly coincide compared with the other two. This indicates that this solution behavior matches the characteristics of a Newtonian fluid.

However, a thixotropic ring is formed at the high rates, which indicates that there is a strong network structure in the solution at higher shear rates and the rate of structural destruction for the polymer solution will be greater than its structural recovery rate.

Furthermore, the area of the hysteresis loop clearly increases with an increase in concentration, suggesting that the strength of the fluid’s network structure can be increased further by simply increasing the HELV concentration. From a microscopic point of view, this interaction is attributable to the transition of the fluid’s system from a loose structure to a dense and entangled network, which further implies that an increase in polymer concentration will improve the structural strength of the polymer system overall.

### 3.8. Viscoelastic Heat-Resistance Measurement

A stress sweep test confirmed that an applied amplitude of 0.1% was within the linear viscoelastic limit. In light of this, a sinusoidal amplitude of 0.1% was applied at an oscillation frequency of 1 Hz in order to determine the temperature limits of the study solution’s elastic and viscous regimes. Figure 12 presents the results for *G′* and *G″* as a function of temperature for tested fluids. It can be seen that the elastic regimes of the fluids are more dominant than their viscous regimes when the temperature is raised from 25 to 80 °C. Moreover, it was found that increasing the polymer concentration from 0.1 to 0.3 wt% brings about a dramatic increase in viscoelasticity.

In order to measure the viscoelastic heat-resistance of the study’s fluids further, samples were put under a temperature of 80 °C and tested again. Results are presented in Figure 13. It is found that the elastic regimes are more dominant than the viscous regimes when the polymer concentration is 0.2 or 0.3 wt% HELV. However, the values for the elastic regimes are found to be less than those for the viscous regimes when the polymer concentration is at 0.1 wt% HELV. This can be explained by the fact that the concentration will still be below the solution’s critical crosslinking concentration (*C*_cc_) and second critical associating concentration (*C_2_^*^*). As such, the fluid’s spatial network structure will still be loose. To sum up the interaction simply, when the fluid is placed in a test environment, its viscoelasticity will be immediately eliminated, and the fluid will change from one that is elastic to one that is more viscous.

### 3.9. Thermo-Shearing Resistance and Thermodynamic Property

Figure 14a shows the profiles of apparent viscosity versus temperature under 170 s^−1^. Test fluids were prepared with different concentrations of HELV (0.1, 0.2 and 0.3 wt%) and 0.015 wt% SDS in deionized water. It is clear that the viscosity of test fluids decreased with the increase of temperature. Moreover, compared with the viscosity of the fluids at 30 °C, the retention rates of the viscosity of all of the test fluids were 55.1%, 52.3%, and 50.0% at 80 °C, respectively. These results indicate that the test fluids have good temperature resistance, indicating that the introduction of hydrophobic long chains and a benzene ring would prevent the bending of the polymer backbone at thermo-shear. Meanwhile, it was found that the relationship between the viscosity and temperature can be expressed by an Arrhenius-type equation as shown in Equation (4) [23,24]:(4)η=Aexp(EaRT)
where *η* is the viscosity, *E*_a_ is the activation energy in J/mol, *R* is the universal gas constant in 8.314 J/(mol·K), *T* is the absolute temperature in Kelvin, and *A* is a characteristic constant of the material. According to the theory of reaction kinetics, in the process of elementary reaction, only the collisions among activated molecules can lead to effective reactions. Among the parameters of Equation (4), *E*_a_ indicates the difference between the average energy of activated molecules and that of reactant molecules. Equation (4) can be modified as a form of the relationship between the natural logarithm of η and the reciprocal of *T*, as shown in Equation (5):(5)lnη=EaRT+lnA

Equation (5) shows a straight line with a slope of *E*_a_/*R* and an intercept of ln*A*. The relationship of the Arrhenius-type equation where lnη responds as a function of 1/*T* with different test fluids is shown in Figure 14b, and Equation (5) can be described as Equations (6)–(8) based on the linear regression and calculation:(6)lnη=1355.79T−0.41486
(7)lnη=1348.12T−0.91584
(8)lnη=1684.45T−0.80882

Figure 14 shows plots of the logarithm for viscosity versus the reciprocal temperature. The linear plots in the figure show data obtained from the heat-shearing resistance tests, which are in agreement with the Arrhenius equation. The values of the activation energy were calculated using the slope value (*E*_a_/*RT*) from the Arrhenius-type equation; the results are in Table 3. The test fluids exhibited lower values of activation energy, 11.208, 11.272 and 14.004 kJ/mol and the activation energy of different test fluids increased with the increase in concentration of HELV. This is because viscosity is a weak function of temperature (and hence low value of activation energy) in contrast to the living polymer solutions, for typical polymer solutions [23]. This strong dependence of viscosity on temperature has been interpreted to occur as a consequence of the decrease in the average length upon increasing the temperature. Thus, the low activation energy values resulted in greater thermal stability of the fluid and a more stable network structure compared to the others.

### 3.10. Dynamic Sand Suspension

#### 3.10.1. Proppant Migration

The positions of the proppants in the fracturing fluid system are shown in Figure 15a,b. For this, the proppants were added to a flat plate when the fracturing fluid was pumped into cracks with a displacement of 1.0 m^3^/min. The transport velocity of these proppants in the horizontal direction was very fast, and thus settlement velocity was correspondingly slow, indicating that the study’s solution exhibits excellent sand carrying capacity. 

#### 3.10.2. Proppant Placement

The positioning of the proppants in water, a 0.3 wt% HPG solution, and a 0.3 wt% HELV solution, were compared. The results are shown in Figure 16a–d. It can be clearly seen that the proppants have subsided almost completely at the seam of the crack when proppants were suspended in water with a displacement of 1.0 m^3^/min. The position of the proppants in the crack was investigated further after mixing with 0.3 wt% HPG solution as well as 0.3 wt% HELV solution. Here the proppants in the HPG solution were found primarily at the crack’s inlet as well as scattered about the middle of the crack. The proppants in the HELV solution, however, were found distributed evenly throughout the crack. As such, it is reasonable to assume that proppant suspension in the HELV solution is superior to that of the HPG solution or water. This indicates that proppants will be more easily transported into deep cracks when using the study’s HELV solution. Thus, it can be concluded that effective fracture length can be increased in hydraulic fracturing operations when the study’s HELV solution is used to carry proppants.

### 3.11. Mechanism Analysis for Superior Proppant Carrying Capacity

The dynamic sand suspension testing clearly demonstrated that HELV exhibits superior proppant carrying capability when compared with the same concentration of HPG. This can be attributed to the HELV fluid’s reduced shear rate at the plate’s center due to the non-uniform nature of the solution’s Poiseulle flow [15]. Consequently, the proppant in this region will be suspended in the HELV solution for a longer period than for HPG, enabling particle migration for a much longer time.

In the near-wall region, where shear rate is high, the higher elasticity of the HELV solution likely plays an important role in sand suspension. The effect of elasticity and viscosity on particle settling can be leveraged to estimate settling velocity. For a single particle in an inelastic fluid, settling velocity will be governed by Stokes’ law [15]:(9)$$v_{0}=\frac{1}{18}\frac{\left(\rho_{p}-\rho_{m}\right)gD^{2}}{\eta }$$
where *ρ_p_* is the particle density (g/cm^3^), *ρ_m_* is the medium density (g/cm^3^), *g* is the gravitation constant (mm/s^2^), *D* is the particle diameter (mm), and η is the medium viscosity (mPa·s). For sand slurries with a non-negligible particle–particle interaction, Stokes’ law must be modified to account for said particle–particle interaction. A classical empirical approach for achieving this is the Richardson–Zaki equation [25]:(10)$$v=v_{0}{\left(1-\varphi \right)}^{n}$$
where *v*_0_ is the velocity of a single particle, as predicted by Stokes’ law, *ϕ* is the particle volume fraction, and *n* is the power law index, which is notably dependent on the Reynolds number for falling particles $\frac{\rho_{p}v_{0}D^{2}}{\eta }$, and equals:
*n* = 4.65*Re* < 0.2*n* = 4.4*Re*^−0.03^0.2 < *Re* <1*n* = 4.4*Re*^−0.1^1 < *Re* < 500*n* = 2.4500 < *Re*

For 0.3 wt% HELV, when using a measured viscosity of 26 mPa·s, a sand density of 1.8 g/cm^3^, a particle diameter of 0.63 mm for 20/40 mesh sand, a particle volume fraction ϕ of 0.06, and an *n* = 4.65 (*Re* = 0.034 is far less than 0.2), a settling velocity of approximately 4.98 mm/s was obtained.

When only considering the effect that viscosity has on particle settling, the velocity is estimated at 4.98 mm/s using Equations (9) and (10) with a fluid viscosity of 26 mPa·s at 170 s^−1^. However, the results for the 0.3 wt% HELV solution exhibited significant elasticity, as shown in Figure 7, Figure 8, Figure 11, Figure 12 and Figure 13, something that must be taken into account when it comes to calculating settling velocity. For investigating viscoelastic shear-thinning fluids specifically, the ratio between viscosity and elasticity therein (λ*_e/v_*) can be calculated using Equation (11):(11)$$\lambda_{e/v}=\frac{0.5c\tau N_{1}(\overset{•}{r})}{\eta (\overset{•}{r})}$$
where *c* is the constant for a given fluid, τ is fluid relaxation time, $N_{1}(\overset{•}{r})$ is the first normal stress, and $\eta (\overset{•}{r})$ is the shear viscosity [16]. It is found that the fluid relaxation time (τ) for the 0.3 wt% HELV solution is 0.6 s, $\eta (\overset{•}{r})=1.317{\overset{•}{r}}^{-0.599}$ Pa·s (Figure 5), $N_{1}(\overset{•}{r})=0.0371{\overset{•}{r}}^{1.14}$ Pa (Figure 7), and $\overset{•}{r}=170s^{-1}$. Assuming *c* as being on the order of 0.1, following the methodology of similar work in the field [16], an equation of $\lambda_{e/v}\sim 0.00087{\overset{•}{r}}^{1.6}=6.39$ is obtained. 

Thus, the settling velocity will be 1/7.39 times the value estimated for shear viscosity, equal to 0.674 mm/s here. Based on these calculations, it can be concluded that the increased elasticity of HELV contributes significantly to improved proppant transport capacity.

### 3.12. Field-Scale Friction Reduction Evaluation

The relationship between the shear rate and drag reduction rate was investigated using a reduction performance evaluation device developed for the study. The device was built to test for a pipe diameter of 6 mm. Results for testing therein are presented in Figure 17 and Figure 18.

#### 3.12.1. Drag Reduction Rate of HELV and HPG

As seen in Figure 17, the drag reduction rate for the study’s HELV solution is superior to that for HPG under variable shear rates between 1000 and 7000 s^−1^. The maximum drag reduction rate for the HELV mixture was found to be 70.57%, which was higher than the results for HPG at 68.68%. In addition, it is noticeable that the drag reduction rate for both HELV and HPG initially increased, but then stabilized with increasing shear rate. This is because the fluids will be in a horizontal flow state when shear rate is still low. Under a condition of laminar flow, the drag reducing agents become disordered, and the fluid interface curvatures for both HELV and HPG exhibit smaller unit lengths [26]. Overall, however, it was found that the drag reducing agents had little effect on general drag reduction.

However, a fracturing fluid will be turbulent when shear rate is high. Under this turbulent state, a fluid’s interface curvature along a length of pipe will be large. The molecules in the drag reducer will be fully dispersed and arrange themselves linearly, which reduces the difference in velocity on the interface [27]. As such, a drag reducer can be applied to stimulated reservoir volume with large displacement and large liquid.

#### 3.12.2. Drag Reduction Rate of HELV with Different Concentrations

As seen in Figure 18, the drag reduction rates for the HELV solution with concentrations of 0.05, 0.10, and 0.15 wt% reach a maximum of 71.24%, 70.57% and 68.61%, respectively. This demonstrates that the drag reduction properties of the HELV fluids will continuously reduce as concentration increases. The possible reason for this is that drag reducers were saturated in unit volume, indicating that intermolecular force stabilized with increasing amounts of HELV. Simultaneously, the viscous force between the fluids themselves and the tube wall will increase as the concentration of the drag reducer continues to rise. Thus, drag reduction rate will gradually decrease overall with continuous increases in HELV concentration.

### 3.13. Breaker

Past work in the field has shown that linear polyacrylamide gels can be destroyed by an oxidizer, such as ammonium persulfate (APS) or potassium persulfate [27]. With this in mind, APS was used to break down HELV solutions for this experiment. Figure 19 presents the viscosity profile of a fluid containing 0.3 wt% HELV and 0.015 wt% SDS with different concentrations of APS added at 80 °C. It is found that, when using APS, the fracturing fluid is broken down completely, and the viscosity of the breaking fluid will be less than 5 mPa·s when the amount of APS added is up to 0.08 wt% and when it is added within 3 h. The resulting fluid is transparent, exhibiting no visible residue.

## 4. Conclusions

This study has delved into issues regarding fracturing fluid viscosity, improving proppant suspension capacity, as well as pipeline and liquid friction. A fracturing fluid was presented as a novel and useful tool for utilization in large-scale fracking. From the results collected from this study’s series of laboratory experiments, the following conclusions can be drawn:(1)The polymer, HELV, was synthesized by radical polymerization and characterized by ^1^H NMR. The tests of critical associating concentration and SEM suggested that there is a multilayered and robust network structure in the polymer solution.(2)It was observed that there is an excellent elasticity in the polymer solution by the first normal stress difference, viscoelasticity, and thixotropy. The critical crosslinker concentration of polymer with sodium dodecyl sulfate and its interaction mechanism were investigated.(3)According to reaction kinetics, the supramolecular structure had the lowest activation energy, stable network structure, and greater thermal stability.(4)The polymer was employed in the fracturing fluid due to its excellent elasticity using the intermolecular forces, which showed superior sand suspension capacity by dynamic sand suspension measurement. Meanwhile, a theoretical analysis is proposed as to why polymer solution has excellent suspension and drag reduction properties.(5)When encapsulated, APS was added at the amount of approximately 0.08 wt% and the viscosity of the breaker was less than 5 mPa·s, making the fluid transparent without visible residue. Therefore, this polymer could be an alternative in many fields, especially in fracking, which is significant for the development of oil and gas resources in deep wells.

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
