# Peer review of "Preparation of a Novel Fracturing Fluid System with Excellent Elasticity and Low Friction"

_polymers, 2019, doi:10.3390/polym11101539_

Round 1

Reviewer 1 Report

The article is valuable and interesting. The only suggestions are connected with some figures.

Fig. 18b can be skipped because it presents nothing interesting and valuable. Fig. 15, 16 are not clear. I would suggest improving the brightness and contrast. In addition, Figures 15 a-d can be combined into one figure (one common description). The same suggestion applies to Figure 16. Fig. 17 is double. The correction of the numbering of figures is needed.

Author Response

Reviewer #1:

Comment 1: The article is valuable and interesting. The only suggestions are connected with some figures.

Reply: First of all, thank you very much for your positive comments on my manuscript. We have revised the manuscript as your suggestion. A point by point response to your comments have been listed below.

Some other problems

We have revised the manuscript point-to-point as the reviewers suggested. It was embarrassed to find many errors and now we have improved them.

Question 1): Fig. 18b can be skipped because it presents nothing interesting and valuable. Fig. 15, 16 are not clear. I would suggest improving the brightness and contrast. In addition, Figures 15 a-d can be combined into one figure (one common description). The same suggestion applies to Figure 16. Fig. 17 is double. The correction of the numbering of figures is needed.

Answer 1): First of all, thank you very much for your valuable advice. As a suggestion, we have revised the error in Figure 15, 16, 17, and 18 in the manuscript, which are marked in blue.

Figure 15. The diagram for proppant migration: (a) 20/40 mesh; (b) 30/50 mesh.

Figure 16. The diagram for proppant placement: (a) 20/40 mesh proppant in water; (b) 20/40 mesh proppant in 0.3 wt% HPG solution; (c) 20/40 mesh proppant in 0.3 wt% HELV solution; (d) 30/50 mesh proppant in 0.3 wt% HELV solution.

3.11. Field-scale friction reduction evaluation

The relationship between the shear rate and drag reduction rate was investigated using a reduction performance evaluation device developed for the study. The device was built to test for a pipe diameter of 6 mm. Results for testing therein are presented in Figure 17 and Figure 18.

As seen in Figure 17, the drag reduction rate for the study’s HELV solution is superior to that for HPG under variable shear rates between 1000 s-1 and 7000 s-1. The maximum drag reduction rate for the HELV mixture was found to be 70.57%, which was higher than the results for HPG at 68.68%. In addition, it is noticeable that the drag reduction rate for both HELV and HPG initially increased, but then stabilized with increasing shear rate. This is because the fluids will be in a horizontal flow state when shear rate is still low. Under a condition of laminar flow, the drag reducing agents became disordered, and the fluid interface curvatures for both HELV and HPG exhibited smaller unit lengths [1]. Overall, however, it was found that the drag reducing agents had little effect on general drag reduction.

However, a fracturing fluid will be turbulent when shear rate is high. Under this turbulent state, a fluid’s interface curvature along a length of pipe will be large. The molecules in the drag reducer will be fully dispersed and arrange themselves in a linearly, which reduces the difference in velocity on the interface [2]. As such, a drag reducer can be applied to stimulated reservoir volume with large displacement and large liquid.

Figure 17. The variation of drag reduction rate between HELV and HPG at different shear rates.

As seen in Figure 18, the drag reduction rates for the HELV solution with concentrations of 0.05 wt%, 0.10 wt%, and 0.15 wt% reach a maximum of 71.24%, 70.57%, and 68.61%, respectively. This demonstrates that the drag reduction properties of the HELV fluids will continuously reduce as concentration increases. The possible reason for this is that drag reducers were saturated in unit volume, indicating that intermolecular force stabilized with increasing amounts of HELV. Simultaneously, the viscous force between the fluids themselves and the tube wall will increase as the concentration of the drag reducer continues to rise. Thus, drag reduction rate will gradually decrease overall with continuous increases in HELV concentration.

Figure 18. The variation of drag reduction rate between HELV concentrations at different shear rates.

3.12. Breaker

Past work in the field has shown that linear polyacrylamide gels can be destroyed by an oxidizer, such as ammonium persulfate (APS) or potassium persulfate [2]. With this in mind, APS was used to break down the HELV solutions for this experiment. Figure 19 presents the viscosity profile of a fluid containing 0.3 wt% HELV and 0.015 wt% SDS with different concentrations of APS added at 80 °C. It is found that, when using APS, the fracturing fluid is broken down completely, and that the viscosity of the breaking fluid will be less than 5 mPa∙s when the amount of APS added is up to 0.08 wt% and when it is added within 3 h. The resulting fluid is transparent, exhibiting no visible residue.

Figure 19. The effect of the amount of APS on the breaking fluid viscosity.

Reviewer 2 Report

This paper was delved into issues regarding fracturing fluid viscosity, improving proppant suspension capacity, as well as pipeline and liquid friction. A fracturing fluid was presented as a novel and useful tool for utilization in large-scale fracking.

The originality the concepts, the significance and the methods are good. The completeness and the organization of manuscript of the paper are good. In my opinion the technical treatment is plausible and free of technical errors.

Here are some comments:

Lines 78-86: I think it would be worth giving the CAS numbers of the used chemicals Line 87: Correct the formatting of the subsection (align left, italics) Line 88: What are the "Appropriate amounts"? Line 155: Use full name – Equation. Line 294 and 297: Why reference “[19]” are marked in blue? Check in Polymers template how to describe Figures 15, 16, and 17. "Authors contributions" is missing.

Author Response

Reviewer #2:

Comment 1: This paper was delved into issues regarding fracturing fluid viscosity, improving proppant suspension capacity, as well as pipeline and liquid friction. A fracturing fluid was presented as a novel and useful tool for utilization in large-scale fracking.

The originality the concepts, the significance and the methods are good. The completeness and the organization of manuscript of the paper are good. In my opinion the technical treatment is plausible and free of technical errors.

Reply: First of all, thank you very much for your positive comments on my manuscript. We have revised the manuscript as your suggestion. A point by point response to your comments have been listed below.

Question 1): Lines 78-86: I think it would be worth giving the CAS numbers of the used chemicals Line 87: Correct the formatting of the subsection (align left, italics) Line 88: What are the "Appropriate amounts"? Line 155: Use full name – Equation. Line 294 and 297: Why reference “[19]” are marked in blue? Check in Polymers template how to describe Figures 15, 16, and 17. "Authors contributions" is missing.

Answer 1): Dear reviewer, your suggestion is reasonable that it is significant for the promotion of my paper. We have added the CAS numbers of the used chemicals and correct the formatting of the subsection Line 88. In addition, we have supplied the specific amounts of the used chemicals. And we have revised all the abbreviated names of “Equation” to its full names in the manuscript. Moreover, it is embarrassing that the reference “[19]” are marked in blue in the manuscript. We have revised it from blue to black. And we have re-described the Figures 15, 16, and 17 according the Polymers template. The "Authors contributions" is added in the manuscript.

Acrylamide (AM, CAS: 79-06-1), acrylic acid (AA, CAS: 79-10-7), sodium hydroxide (NaOH, CAS: 1310-73-2), sodium dodecyl sulfate (SDS, CAS: 751-21-3), 2,2'-azobis (2-methylpropionamide) dihydrochloride (V50, CAS: 2997-92-4) were all purchased from Chengdu Kelong Chemical Reagents Corporation (Chendu, P.R. China). 4-isopropenylcarbamoyl-benzene sulfonic acid (AMBS), N-(3-methacrylamidopropyl)-N,N-dimethyldodecan-1-aminium (DM-12) were prepared in our lab as functional monomer. All chemicals and reagents were utilized without further purification. Ammonium persulfate (APS, CAS: 7727-54-0), hydroxypropyl guar gum (HPG) and proppant were applied by GuangHan Innovative Technology Development Institute. Deionized water was obtained from a water purification system. All chemicals were analytical reagent grade and were utilized without further purification.

2.2. Synthesis of HELV

Appropriate amounts of acrylamide (12.0 g), acrylic acid (3.0 g), 4-isopropenylcarbamoyl-benzene sulfonic acid (0.4 g), and N-(3-methacrylamidopropyl)-N,N-dimethyldodecan-1-aminium (0.4 g) were placed in deionized water,

A 6 mm pipeline was used to test the relationship between pressure drop and flow rate. Specifically, drag reduction is defined by (Equation. 1), as follows

Consistency index (κ) values as well as flow behavior index (η) values for the tested fluids were obtained by curve fitting the power law model to the study’s shear stress versus shear rate data (Equation. 2):

where η is the viscosity, Ea is the activation energy in J/mol, R is the universal gas constant in 8.314 J/(mol·K), T is the absolute temperature in Kelvin, and A is a characteristic constant of the material. According to the theory of the reaction kinetics, in the process of elementary reaction, only the collisions among activated molecules can lead to effective reactions. Among the parameters of Equation. 4, Ea indicates the difference between the average energy of activated molecules and that of reactant molecules. Equation. 4 can be modified as a form of the relationship between the natural logarithm of η and the reciprocal of T, as shown in Equation. 5:

Equation 5 shows a straight line with a slope of Ea/R and an intercept of ln A. The relationship of Arrhenius-type that ln η responds as a function of 1/T with the different test fluids is shown in Figure 14b, and Equation. 5 can be described as Equation. 6, Equation. 7, Equation. 8 based on the linear regression and calculation:

When only considering the effect that viscosity has on particle settling, the velocity is estimated at 4.98 mm/s using Equation. 9 and Equation. 10 with a fluid viscosity of 26 mPa∙s at 170 s-1. However, the results for the 0.3 wt% HELV solution exhibited significant elasticity, as shown in Figure 7, Figure 8, Figure 11, Figure 12, and Figure 13, something that must be taken into account when it comes to calculating settling velocity. For investigating viscoelastic shear-thinning fluids specifically, the ratio between viscosity and elasticity therein (λe/v) can be calculated using Equation. 11:

The concentration at which all of the relationships curves are found to have the same value for tan δ will be the solution’s critical crosslinking concentration (Ccc) [3]. For this study’s testing for Ccc, the results are presented in Figure 10.

3.10. Dynamic sand suspension

3.10.1. Proppant migration

The positions of the proppants in the fracturing fluid system are shown in Figures 15a-b. For this, the proppants were added to a flat plate when the fracturing fluid was pumped into cracks with a displacement of 1.0 m3/min. The transport velocity of these proppants in the horizontal direction was very fast, and thus settlement velocity was correspondingly slow, indicating that the study’s solution exhibits excellent sand carrying capacity.

Figure 15. The diagram for proppant migration: (a) 20/40 mesh; (b) 30/50 mesh.

3.10.2. Proppant placement

The positioning of the proppants in water, a 0.3 wt% HPG solution, and a 0.3 wt% HELV solution were compared. The results are shown in Figures 16a-d. It can be clearly seen that the proppants have subsided almost completely at the seam of the crack when proppants were suspended in water with a displacement 1.0 m3/min. The position of the proppants in the crack was investigated further after mixing with the 0.3 wt% HPG solution as well as the 0.3 wt% HELV solution. Here the proppants in the HPG solution were found primarily at the crack’s inlet as well as scattered about the middle of the crack. The proppants in the HELV solution, however, were found distributed evenly throughout the crack. As such, it is reasonable to assume that proppant suspension in the HELV solution is superior to that of the HPG solution or water. This indicates that proppants will be more easily transported into deep cracks when using the study’s HELV solution. Thus, it can be concluded that effective fracture length can be increased in hydraulic fracturing operations when the study’s HELV solution is used to carry proppants.

Figure 16. The diagram for proppant placement: (a) 20/40 mesh proppant in water; (b) 20/40 mesh proppant in 0.3 wt% HPG solution; (c) 20/40 mesh proppant in 0.3 wt% HELV solution; (d) 30/50 mesh proppant in 0.3 wt% HELV solution.

3.12. Field-scale friction reduction evaluation

The relationship between the shear rate and drag reduction rate was investigated using a reduction performance evaluation device developed for the study. The device was built to test for a pipe diameter of 6 mm. Results for testing therein are presented in Figure 17 and Figure 18.

3.12.1. Drag reduction rate of HELV and HPG

As seen in Figure 17, the drag reduction rate for the study’s HELV solution is superior to that for HPG under variable shear rates between 1000 s-1 and 7000 s-1. The maximum drag reduction rate for the HELV mixture was found to be 70.57%, which was higher than the results for HPG at 68.68%. In addition, it is noticeable that the drag reduction rate for both HELV and HPG initially increased, but then stabilized with increasing shear rate. This is because the fluids will be in a horizontal flow state when shear rate is still low. Under a condition of laminar flow, the drag reducing agents became disordered, and the fluid interface curvatures for both HELV and HPG exhibited smaller unit lengths. Overall, however, it was found that the drag reducing agents had little effect on general drag reduction.

However, a fracturing fluid will be turbulent when shear rate is high. Under this turbulent state, a fluid’s interface curvature along a length of pipe will be large. The molecules in the drag reducer will be fully dispersed and arrange themselves in a linearly, which reduces the difference in velocity on the interface. As such, a drag reducer can be applied to stimulated reservoir volume with large displacement and large liquid.

Figure 17. The variation of drag reduction rate between HELV and HPG at different shear rates.

3.12.2. Drag reduction rate of HELV with diferent concentration

As seen in Figure 18, the drag reduction rates for the HELV solution with concentrations of 0.05 wt%, 0.10 wt%, and 0.15 wt% reach a maximum of 71.24%, 70.57%, and 68.61%, respectively. This demonstrates that the drag reduction properties of the HELV fluids will continuously reduce as concentration increases. The possible reason for this is that drag reducers were saturated in unit volume, indicating that intermolecular force stabilized with increasing amounts of HELV. Simultaneously, the viscous force between the fluids themselves and the tube wall will increase as the concentration of the drag reducer continues to rise. Thus, drag reduction rate will gradually decrease overall with continuous increases in HELV concentration.

Figure 18. The variation of drag reduction rate between HELV concentrations at different shear rates.

Author Contributions: Data curation, Yang Zhang, Tao Xu and Anqi Du; Funding acquisition, Jincheng Mao and Jinzhou Zhao; Investigation, Tao Xu; Methodology, Zhaoyang Zhang, Wenlong Zhang and Shaoyun Ma; Project administration, Jincheng Mao and Jinzhou Zhao; Writing – original draft, Yang Zhang; Writing – review & editing, Yang Zhang, Jincheng Mao, Anqi Du, Zhaoyang Zhang, Wenlong Zhang and Shaoyun Ma.

Funding: The research is partly supported by Sichuan Youth Science & Technology Foundation (2017JQ0010), National High Technology Research & Development Program (2016ZX05053, 2016ZX05014-005-007), Key Fund Project of Educational Commission of Sichuan Province (16CZ0008), Explorative Project Fund (G201601) of State Key Laboratory of Oil and Gas Reservoir Geology and Exploitation (Southwest Petroleum University), China Postdoctoral Science Foundation (2019M650250), and the National Natural Science Foundation of China (Grant Nos. 41902303).
